



# A strontium isoscape of northern Australia

Patrice de Caritat[1], Anthony Dosseto[2], Florian Dux[2]

[1]Geoscience Australia, GPO Box 378, Canberra ACT 2601, Australia
[2]Wollongong Isotope Geochronology Laboratory, School of Earth, Atmospheric and Life Sciences, University of Wollongong, Wollongong NSW 2522, Australia

*Correspondence to*: Patrice de Caritat (Patrice.deCaritat@ga.gov.au)

**Abstract (384 words)**

Strontium isotopes ($^{87}Sr/^{86}Sr$) are useful in the Earth sciences as well as in forensic, archaeological, palaeontological, and ecological sciences. As very few large-scale Sr isoscapes exist in Australia, we have identified an opportunity to determine $^{87}Sr/^{86}Sr$ ratios on archive fluvial sediment samples from the low-density National Geochemical Survey of Australia (www.ga.gov.au/ngsa; last access: 15 December 2022). The present study targeted the northern parts of Western Australia, the Northern Territory and Queensland, north of 21.5 °S. The samples were taken mostly from a depth of ~60-80 cm in floodplain deposits at or near the outlet of large catchments (drainage basins). A coarse (< 2 mm) grain-size fraction was air-dried, sieved, milled then digested (hydrofluoric acid + nitric acid followed by aqua regia) to release total Sr. The Sr was then separated by chromatography and the $^{87}Sr/^{86}Sr$ ratio determined by multicollector-inductively coupled plasma mass spectrometry. Preliminary results demonstrate a wide range of Sr isotopic values (0.7048 to 1.0330) over the survey area, reflecting a large diversity of source rock lithologies, geological processes and bedrock ages. Spatial distribution of $^{87}Sr/^{86}Sr$ shows coherent (multi-point anomalies and smooth gradients), large-scale (> 100 km) patterns that appear to be broadly consistent with surface geology, regolith/soil type, and/or nearby outcropping bedrock. For instance, the extensive black clay soils of the Barkly Tableland define a > 500 km-long northwest-southeast-trending unradiogenic anomaly ($^{87}Sr/^{86}Sr$ < 0.7182). Where carbonate or mafic igneous rocks dominate, a low to moderate Sr isotope signature is observed. In proximity to the outcropping Proterozoic metamorphic basement of the Tennant, McArthur, Murphy and Mount Isa geological regions, conversely, radiogenic $^{87}Sr/^{86}Sr$ values (> 0.7655) are observed. A potential correlation between mineralisation and elevated $^{87}Sr/^{86}Sr$ values in these regions needs to be investigated in greater detail. Our results to-date indicate that incorporating soil/regolith Sr isotopes in regional, exploratory geoscience investigations can help identify basement rock types under (shallow) cover, constrain surface processes (e.g. weathering, dispersion), and, potentially, recognise components of mineral systems. Furthermore, the resulting Sr isoscape and future models derived therefrom can also be utilised in forensic, archaeological, paleontological and ecological studies that aim to investigate, e.g., past and modern animal (including humans) dietary habits and migrations. The new spatial Sr isotope dataset for the northern Australia region is publicly available (de Caritat et al., 2022a; https://dx.doi.org/10.26186/147473; last access: 15 December 2022).



# 1 Introduction

Strontium isotope ratios ($^{87}$Sr/$^{86}$Sr) can be measured in many geological materials as the trace element strontium (Sr) is relatively abundant and readily substitutes for calcium (Ca) in minerals and organic tissues. The $^{87}$Sr/$^{86}$Sr of a mineral or rock is a function of (1) its initial and unchanging $^{86}$Sr content, (2) its initial rubidium (Rb) content (thus Rb/Sr ratio), and (3) time (e.g. McNutt, 2000). Rubidium substitutes readily for potassium (K) in minerals and is thus relatively common too. As one of the two naturally occurring Rb isotopes, $^{87}$Rb, which accounts to 27.8 % of Rb, decays over time (by emitting a negative beta particle) to stable $^{87}$Sr ($t_{1/2}$ = 49.6 x 10$^9$ years), the $^{87}$Sr/$^{86}$Sr ratio of that material slowly increases with time (e.g. Rotenberg et al., 2012; Nebel and Stammeier, 2018). During geological processes such as mineral dissolution or precipitation, or biological processes such as bone and tooth growth, the $^{87}$Sr/$^{86}$Sr remains constant as there is no fractionation (e.g. Gosz et al., 1983; Nebel and Stammeier, 2018). These characteristics make the Sr isotopic system very useful in the geosciences, where it has been used for decades for instance in the study of

- sediment diagenesis and low-grade metamorphism (e.g. Swart et al., 1987; Schultz et al., 1989; Mountjoy et al., 1992; Schaltegger et al., 1994),

- sediment, dust and soil provenance (e.g. Douglas et al., 1995; Revel-Rolland et al., 2006; Bathgate et al., 2011; Bataille and Bowen, 2012; De Deckker et al., 2014; Jomori et al., 2017; De Deckker, 2020; de Caritat et al., 2022b),

- stratigraphy and reservoir/basin analysis (e.g. Rundberg and Smalley, 1989; Smalley et al., 1989, 1992; McArthur et al., 2012),

- fluid flow (e.g. McNutt et al., 1987; Stueber et al., 1987; Sullivan et al., 1990; Bagheri et al., 2014),

- weathering and pedogenesis (e.g. Åberg et al., 1989; Faure and Felder, 1981; Wickman and Jacks, 1992; Blum et al., 1994; Bullen et al., 1994, 1997; Quade et al., 1995; Probst et al., 2000; Harrington and Herczeg, 2003; Oliver et al., 2003; Green et al., 2004),

- terrestrial ecosystems and catchment processes (e.g. Graustein, 1989; Jacks et al., 1989; Lyons et al., 1995; Négrel and Grosbois, 1999; Négrel and Pauwels, 2004; Gosselin et al., 2004; Chadwick et al, 2009; Hagedorn et al., 2011),

- hydrology and hydrogeology (e.g. Collerson et al., 1988; Andersson et al., 1992, 1994; Douglas et al., 1995; Yang et al., 1996; Grobe et al., 2000; Dogramaci and Herczeg, 2002; Ojiambo et al., 2003; Palmer et al., 2004; Cartwright et al., 2007; Christensen et al., 2018; Shin et al., 2021),

- environment and environmental change (e.g. Andersson et al., 1990; Åberg, 1995; Åberg et al., 1995; Oishi, 2021),

- (palaeo)climatology and palaeogeographic reconstructions (e.g. Blum and Erel, 1995; Wei et al., 2018; Flecker et al., 2002),

- determining the $^{87}$Sr/$^{86}$Sr of seawater through geological time (e.g. Veizer, 1989; Gruszczynski et al., 1992; Denison et al., 1994a, 1994b; Shields, 2007), and

- ore genesis and mineral exploration (e.g. Le Bas et al., 1997; de Caritat et al., 2005; Daneshvar et al., 2020; Wei et al., 2020; Zhao et al., 2021).

Outside of the geosciences, food tracing and provenancing have also been underpinned by the use of Sr isotopes (e.g.
Voerkelius et al., 2010; Di Paola-Naranjo et al., 2011; Vinciguerra et al., 2015; Hoogewerff et al., 2019; Moffat et al., 2020).
Anthropological studies have relied on $^{87}$Sr/$^{86}$Sr isotope ratios to locate archaeological artefacts or reconstruct ancient human
behaviours (e.g. Frei and Frei, 2013; Willmes et al., 2014, 2018; Adams et al., 2019; Pacheco-Forés et al., 2020; Washburn
et al., 2021). Animal migration studies have also relied on Sr isotope data (e.g. Price et al., 2017). More recently, large-scale
compilations and machine-learned predictions of the $^{87}$Sr/$^{86}$Sr variations up to the continental and even global scale have
been proposed (e.g. Bataille et al., 2014, 2018, 2020).

Strontium isotope landscape maps ('isoscapes') provide the fundamental context required for the interpretation of more
detailed scientific research about processes or provenance. Despite the plethora of research using Sr isotopes to address
various scientific questions, very few Sr isoscapes exist in the southern hemisphere, particularly for soils or covering large
swathes of the Earth's surface (see Bataille et al., 2020). Two exceptions to this are (1) the work by Adams et al. (2019),
which reported $^{87}$Sr/$^{86}$Sr in plant, soil, and biota over ~300 000 km$^2$ on the Cape York Peninsula in Australia; and (2) the
~500 000 km$^2$ Sr isoscape of inland southeastern Australia recently published by our team (de Caritat et al., 2022b). The
present study affords an opportunity to further redress this deficiency and will reduce the northern-hemisphere bias in future
global $^{87}$Sr/$^{86}$Sr models. It also pertains to a land surface that has not been rejuvenated by recent glaciation, consisting of over
% regolith or weathered material (Wilford, 2012), and as a result is abundant in minerals such as kaolinite, illite-smectite,
goethite and hematite.

## 2 Setting

The study area in northern Australia stretches across Western Australia, the Northern Territory and Queensland (hereafter
abbreviated to WA, NT and Qld, respectively) north of 21.5 °S (Figure 1). It is surrounded by the Indian Ocean to the west,
the Timor and Arafura Seas and Gulf of Carpentaria to the north, and the Coral Sea (Pacific Ocean) to the east. The main
climate zones in the area are described as 'Hot humid summer' in the north and along the coast, and 'Warm humid summer'
further south and inland (BOM, 2022a). The vegetation zones are dominated by 'Tropical savanna' and 'Hot grassland with
winter drought' (BOM, 2022b). The 10-year (1996-2005) average minimum and maximum temperatures mostly range 15-24
°C and 27-36 °C, respectively (BOM, 2022c). Average annual rainfall over the four-year period to November 2009 (when
the bulk of the sampling was completed) mostly ranges 400-1500 mm/yr and is strongly seasonal (summer rain) (BOM,
2022d). Physiographically the study area includes, from west to east, the Kimberley, North Australian Plateaus, Barkly-
Tanami Plains, Carpentaria Fall, Carpentaria Lowlands, Peninsular Uplands, and Burdekin Uplands Provinces (Pain et al.,
2011). Topographic altitude ranges from sea level to 1622 m above sea level (asl) at Mt Bartle Frere (Qld's highest point,
located just south of Cairns), and the mean altitude is ~260 m asl (GA, 2008).

The soil types encountered in the study area are, according to the Australian Soil Classification scheme (Isbell et al., 2021;
ASRIS, 2022a), most commonly vertosol, kandosol or tenosol (~24 % of the sample sites each), followed by rudosol,

hydrosol or sodosol (7-9 %), and the rarer calcarosol, chromosol, dermosol or podosol (< 2 %). The major river basins that divide the area belong, from west to east, to the Pilbara, Sandy Desert-Mackay, Fitzroy, North Kimberley, Ord, Victoria, Tennant, Daly, Darwin, Melville, Alligator, Roper, Arnhem, McArthur, Channel Country, Burketown, Leichhardt, Flinders, Gilbert, Weipa, Mitchell, Princess Charlotte Bay, Burdekin, Barron, Cooper Creek, and Whitsunday Water Regions (GA,

1997). Landuse over the area is overwhelmingly 'Grazing natural vegetation', followed by 'Other protected areas (inc. indigenous uses)', then 'Nature conservation' (ASRIS, 2022b).

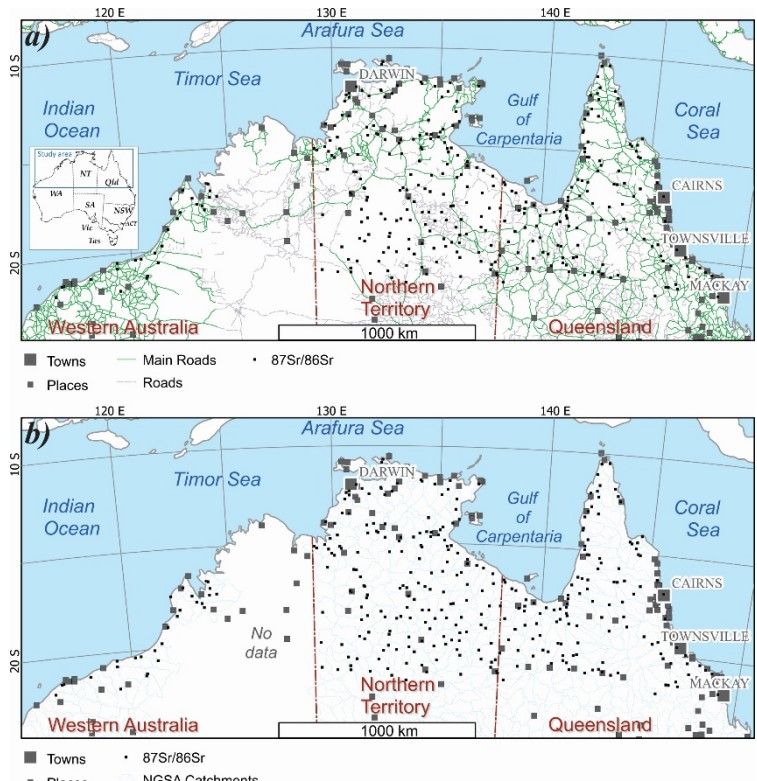

**Figure 1. The northern Australia Sr isotope study area (see inset for location: WA – Western Australia, NT – Northern Territory, Qld – Queensland, NSW – New South Wales, ACT – Australian Capital Territory, Vic – Victoria, Tas – Tasmania, and SA – South**
**Australia) and National Geochemical Survey of Australia (NGSA) and Northern Australia Geochemical Survey (NAGS) 87Sr/86Sr sample locations (small black squares) shown with towns (large grey squares) and places (medium grey squares) as well as (a) main roads (green solid lines), (secondary) roads (grey dashed lines); and (b) NGSA catchment boundaries (medium blue). Map projection: Albers equal area.**

The main geological region groups (Blake and Kilgour, 1998) over the study area are, from west to east, the Pilbara,
Canning, Kimberley, Bonaparte, Ord, Victoria River, Pine Creek, Victoria, Wiso, Arunta, Tennant, Money, McArthur, Arafura, Georgina, Isa, Carpentaria, Eromanga, Georgetown-Coen, Quinkan, and Paleozoic-Qld groups (see Supplement). The higher-level geological region domains (Blake and Kilgour, 1998) over the study area are, from west to east, the Pilbara, Paleozoic, North-Australian Craton, Proterozoic, and Meso-Cenozoic domains (see Supplement). Thus, the study area displays a complex and protracted geological history spanning over 3.6 Gy (Ollier, 1988; Braun et al., 1998; Betts et al.,





2002; Blewett, 2012; Withnall et al., 2013; Ahmad and Scrimgeour, 2013), with two cratonic nuclei in the west and centre (the Archean Pilbara Craton and Paleo- to Mesoproterozoic North-Australian Craton) flanked by younger Proterozoic and Paleozoic orogenic belts and basins, including the Phanerozoic Tasmanides to the east. Mesozoic and Cenozoic sedimentary sequences of the Eromanga and Carpentaria basins conceal much of the basement terrain over the eastern third of the study area. The whole region has experienced extensive weathering resulting in a ubiquitous and locally thick regolith mantle.

Figure 2a shows that by far the dominant bedrock unit intersected by the sample sites is 'Sedimentary and low-grade metamorphics' (89 % of sites), followed by the much less frequent 'Felsic igneous intrusive' (4 %), and 'Mafic to ultramafic igneous volcanic' (3 %) lithologies. Representing 2 % or less each are the lithologies 'Medium-grade metamorphics', 'Felsic to mafic igneous volcanic', 'High-grade metamorphics', 'Mafic igneous intrusive', and 'Felsic to intermediate igneous volcanic'. Investigating further the dominant bedrock unit 'Sedimentary and low-grade metamorphics' reveals that it is

comprised of 'Regolith' (79 %), 'Sedimentary siliciclastic' (4 %), 'Feldspar- or lithic-rich arenite to rudite' (2 %), 'Sedimentary carbonate' (2 %), and < 1 % each of 'Argillaceous detrital sediment', 'Metasedimentary siliciclastic', and 'Quartz-rich arenite to rudite'. Further, the 'Regolith' class is reported to consist mainly 'Alluvium' (51 %), 'Sand plain' (9 %), 'Estuarine and delta deposit' (4 %), 'Colluvium' (4 %), 'Calcrete' (3 %), 'Undifferentiated sediment' (3 %), 'Lake deposit' (2 %), and < 2 % each of 'Black soil plain', 'Dune', and 'Ferruginous duricrust'.

The bedrock ages (periods) intersected at the sample sites are overwhelmingly Tertiary-Quaternary (58 %), followed by much less frequent Mesoproterozoic (8 %), and Cenozoic (6 %), Cambrian (5 %), Paleoproterozoic (5 %), Paleoproterozoic-Mesoproterozoic (3 %), Carboniferous-Permian (3 %), Permian (3 %), Devonian-Carboniferous (2 %), Archaean (2 %), and Cambrian-Ordovician (2 %). Representing < 2 % are the periods Devonian, Jurassic-Cretaceous, Ordovician, Silurian-Devonian, Jurassic, Neoproterozoic-Cambrian, and Triassic.

Numerous mineral occurrences are found in northern Australia and Figure 2b shows the most important ones classified as 'Mineral Deposits' (i.e. those with an inferred resource), 'Operating Mines' (currently producing), or 'Developing Projects' (approved but not yet producing). As mineralisation is the end-point of geologically 'unusual' processes taking place regionally in 'mineral systems' (Wyborn et al., 1994), it is useful to investigate if mineral system processes leave a recognisable $^{87}Sr/^{86}Sr$ fingerprint in the country rock and sediment derived therefrom. The study area is particularly well-

endowed in 'Base Metal' deposits (both Pb-Zn- and Cu-dominated, e.g. Mount Isa, McArthur River, George Fisher) and is host to the 'North Australian Zinc Belt', the world's largest Zn-Pb province (Huston et al., 2022). In addition, notable occurrences of 'Battery or Alloy Metal' (Ni, Co, Mn, V, Mo, and Mg, e.g. Spinifex Ridge, Ripon Hills, Julia Creek, Richmond), and 'Precious Metal' (Au, Ag, e.g. Mount Isa, George Fisher) are also found here. Further, 'Other Metal' (Sn, Sb, W, Ta, Nb, e.g. Mount Carbine), 'Rare Earth Element' (e.g. Thunderbird, Nolans Bore), 'Platinum Group Element' (Pt,

Pd, Rh, e.g. Munni Munni), and 'Heavy Mineral Sands' (Thunderbird) occurrences are also catalogued. Thus, the area is significant for underpinning Australia's critical mineral resources supply now and into the future (e.g. DISER, 2022), without which a global transition to a lower-carbon economy will be challenging.





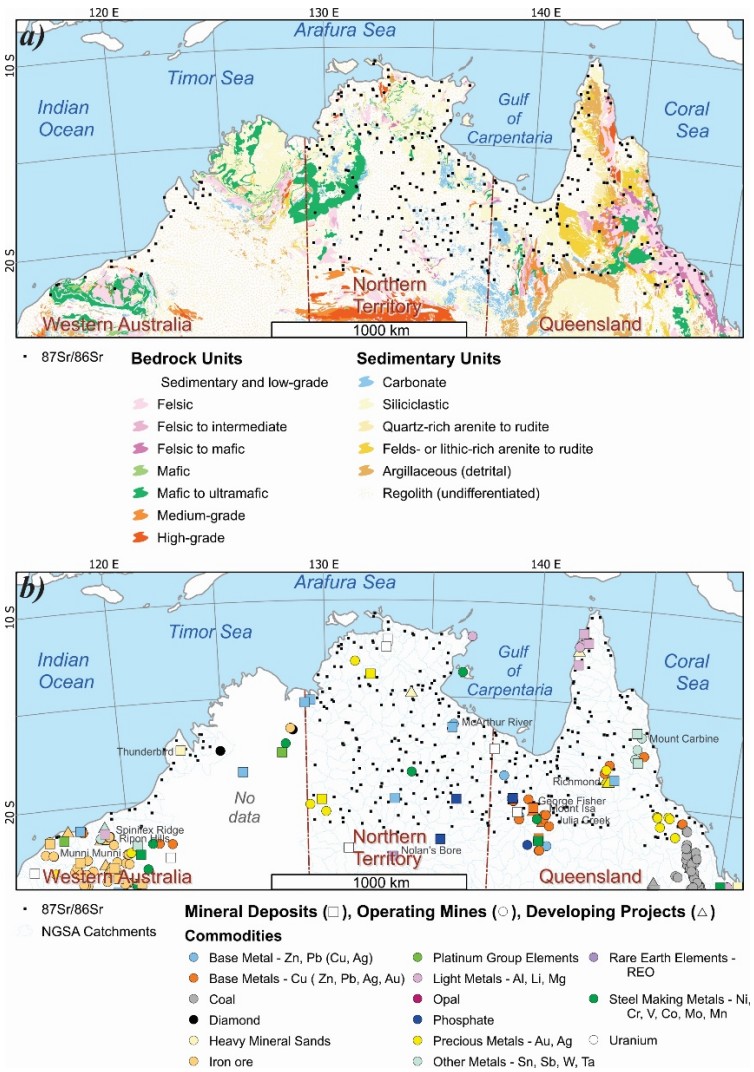

**Figure 2. The northern Australia Sr isotope study area and National Geochemical Survey of Australia (NGSA) and Northern Australia Geochemical Survey (NAGS) $^{87}$Sr/$^{86}$Sr sample locations (small black squares) shown with (a) surface geology from Raymond et al. (2012); and (b) NGSA catchment boundaries (medium blue) and mineral occurrences (GA, 2022a). Map projection: Albers equal area.**

# 3 Material and methods

## 3.1 Material

This study makes use of archive 'catchment outlet sediment' samples collected during the National Geochemical Survey of Australia (NGSA), which covered ~80 % of Australia (de Caritat and Cooper, 2011a, 2016; de Caritat, 2022). The sampling philosophy of the NGSA was to collect naturally mixed and fine-grained fluvial/alluvial sediments from large catchments, thereby obtaining a representative average of the main soil and rock types contributing sediment through weathering. This



allowed an ultralow sampling density (~1 sample per 5200 km$^2$) still being representative of large-scale natural variations (de
Caritat and Cooper, 2011b). Catchment outlet sediments are similar to floodplain sediments in the sense that they are
deposited during receding floodwaters outside the riverbanks, but with the added complexity that, in Australia, many areas
can also experience addition (or loss) of aeolian material. The sampled floodplain geomorphological entities are typically
vegetated and biologically active (plants, worms, ants, etc.) thereby making the collected materials true soils, albeit soils
developed on transported alluvium parent material.

The sampling medium and density were both strategically chosen in the NGSA project to prioritize coverage over resolution.
This was justified by the fact that the NGSA was Australia's first and, to date, only fully integrated, internally consistent
geochemical survey with a truly national scope. In terms of the present study area, it is clear that these choices have
implications on the granularity of the patterns revealed by the Sr isoscape; as the collection of Sr isotope data in Australia
using NGSA samples grows in the future (e.g. de Caritat et al., 2022b, and this contribution), it is hoped the value of
coverage will prevail over a relative low resolution of detailed features.

The NGSA collected samples at two depths, a 'top outlet sediment' (TOS) from a shallow (0.1 m) soil pit approximately 0.8
m x 0.8 m in area, and a 'bottom outlet sediment' (BOS) from a minimum of three auger holes generally drilled within ~10
m of the TOS pit. The auger holes were drilled as deep as possible (to refusal or to maximum depth of 1 m), and the BOS
sample was collected on average from a depth of 0.6 m to 0.8 m from all augered holes. A field manual was compiled to
record all sample collection method details, including site selection (Lech et al., 2007). Sampling for the NGSA took place
between July 2007 and November 2009, and the field data were recorded in Cooper et al. (2010). In the laboratory, the
samples were air dried at 40 °C for a minimum of 48 h (or to constant mass) before being further prepared (see de Caritat et
al., 2009) for the comprehensive geochemical analysis program of the NGSA (see de Caritat et al., 2010). An aliquot of
minimum ~1 g of sample was milled to a fine powder using a carbon steel ring mill or, for a few samples only, an agate
micromill. The main sample type selected for the present Sr isotope study was NGSA BOS < 2 mm in order to be as
representative as possible of the geogenic background unaffected by modern landuse practices and inputs (e.g. fertilizers). A
few NGSA TOS < 2 mm samples, prepared in an identical fashion, were also analysed.

Several additional samples from the Northern Australia Geochemical Survey (NAGS; Bastrakov and Main, 2020) were
included in this project as they provide a higher density coverage over part of the study area. The NAGS project used the
same sampling philosophy and sample collection, preparation and analysis methods as the NGSA, with a higher sampling
density of 1 sample per ~500 km$^2$. The NAGS project collected only TOS (0-0.1 m depth) samples and the NAGS TOS < 2
mm aliquots were prepared and analysed as per the NGSA samples. Sampling for the NAGS took place in May and June
2017.

Overall 326 NGSA BOS < 2 mm, 18 NGSA TOS < 2 mm (including 15 with BOS also analysed), and 28 NAGS TOS
samples, for a total of 372 analyses from 357 samples, were analysed for Sr isotopes as detailed in the Methods Subsection
below. Given that there are ~10 % field duplicates in the NGSA and the smaller NAGS catchments are nested within NGSA



catchments, all those samples originate from within 307 NGSA catchments, which together cover 1.536 million km$^2$ of northern Australia (see Figure 1).

## 3.2 Methods

Samples were prepared and analysed for Sr isotopes ($^{87}$Sr/$^{86}$Sr) at the Wollongong Isotope Geochronology Laboratory (WIGL). Approximately 50 mg of sample was weighed and digested in a 2:1 mixture of hydrofluoric and nitric acids. All reagents used were Seastar Baseline® grade, with Sr concentrations typically < 10 parts per trillion. Following digestion, samples were re-dissolved in aqua regia (twice if needed) in order to eliminate any fluorides, followed by nitric acid twice. Finally, samples were re-dissolved in 2 M nitric acid prior to ion exchange chromatography. Strontium was isolated from the

sample matrix using automated, low-pressure chromatographic system Elemental Scientific prepFAST-MC™ and a 1 mL Sr–Ca column (Eichrom™) (Romaniello et al., 2015). The Sr elutions were re-dissolved in 0.3 M nitric acid. Strontium isotope analysis was performed on a Thermo Scientific Neptune Plus multicollector-inductively coupled plasma-mass spectrometer (MC-ICP-MS) at WIGL. The sample introduction system consists of an ESI Apex-ST PFA MicroFlow nebulizer with an uptake rate of ~0.1 mL min$^{-1}$, an SSI Quartz dual cyclonic spray chamber, jet sample, and X-skimmer

cones. Measurements were performed in low-resolution mode. The instrument was tuned at the start of each session with a 20 parts per billion Sr solution, and sensitivity for $^{88}$Sr was typically around 4 V. Masses 88, 87, 86, 85, 84, and 83 were collected on Faraday cups. Instrumental mass bias was internally corrected using measured $^{87}$Sr/$^{86}$Sr. Masses 85 and 83 were used to correct for the isobaric interference of $^{87}$Rb and $^{86}$Kr, respectively. Maps were prepared using the open software QGis (version 3.16.14–Hannover) and applying an Albers equal area projection. Symbology for displaying $^{87}$Sr/$^{86}$Sr data here was

either point data classified in eight equal quantile classes (12.5 % of the data each; green = low to high = red) at the sampling site, or attributing this same value and colour to the whole catchment from which the outlet sediment comes, reflecting the sampling medium, catchment outlet sediment, being a representative sample of the average materials in the catchment (see Sect. 3.1).

## 3.3 Quality Assessment

National Institute of Standards and Technology (NIST) strontium carbonate isotope Standard Reference Material SRM987 was used as a secondary standard and measured after every five samples to assess accuracy during analysis. Accuracy of the whole procedure was assessed by processing United States Geological Survey (USGS) reference material Basalt from the Columbia River standard BCR-2 (Plumlee, 1998). The mean ± 2se $^{87}$Sr/$^{86}$Sr for BCR-2 in this study is 0.704961 ± 35 ($n$ = 13), within error of the value in Jweda et al. (2016) (0.704500 ± 11). Total procedure blanks ranged between 0.025 and 0.245

ng Sr ($n$ = 12). Twenty field duplicate sample pairs (collected at a median distance of ~80 m from one another on the same landscape unit, see Lech et al., 2007) were analysed for $^{87}$Sr/$^{86}$Sr in the BOS < 2 mm sample, and returned a median relative standard deviation of 0.17 %. The relative standard deviation from field duplicates includes natural variability

(mineralogical/chemical heterogeneity of the alluvial deposit), as well as sample collection, preparation, and analysis uncertainties.

Overall, we feel that the quality of the $^{87}Sr/^{86}Sr$ data presented herein is adequate for the purpose of regional mapping, and that reporting $^{87}Sr/^{86}Sr$ data to the third decimal place with an indicative fourth decimal place is appropriate for this work. This relatively low precision is attributed to heterogeneity of the alluvial deposits, since precision relating to sample preparation and analysis for Sr isotopes is at the fifth decimal place (see results for BCR-2 above).

### 3.4 Data Analysis

Data management was performed using Microsoft Excel®, graphing and visualization using IMDEX ioGas®, and spatial analysis and mapping using QGIS®, an open source geographical information system. For the purposes of generating the Sr isoscape from combined BOS and TOS samples, calculated 'BOS-equivalent' values of TOS were derived from regression analysis (see Sect. 5.2). Catchment-based Sr isoscapes were constructed by assigning to each NGSA catchment the $^{87}Sr/^{86}Sr$ value of its outlet sediment sample, or, if more than one sample analysed per catchment (e.g. field duplicates or higher-

resolution NAGS samples), the mean of those multiple samples. All maps are shown in Albers equal area projection.

### 4 Results

The soil $^{87}Sr/^{86}Sr$ values reported herein range from 0.7048 to 1.0330 (range = 0.3282). The median is 0.7405 and the mean 0.7532 (standard deviation = 0.0480; kurtosis = 8.1602; skewness = 2.4640). Figure 3 illustrates the univariate structure of the new data. Spatially, the $^{87}Sr/^{86}Sr$ values define large-scale, coherent patterns with multi-point low and high regions

(Figure 4). The main high value (radiogenic) regions are found in the central and northern parts of the NT, most of Cape York, as well as along some of the coast including in the WA part of the study. Prominent low $^{87}Sr/^{86}Sr$ value (unradiogenic) regions include a large, central, northwest-southeast trending elongated area in the NT, a smaller and similarly trending area in central Qld, as well as the northernmost and easternmost parts of the Qld study area.


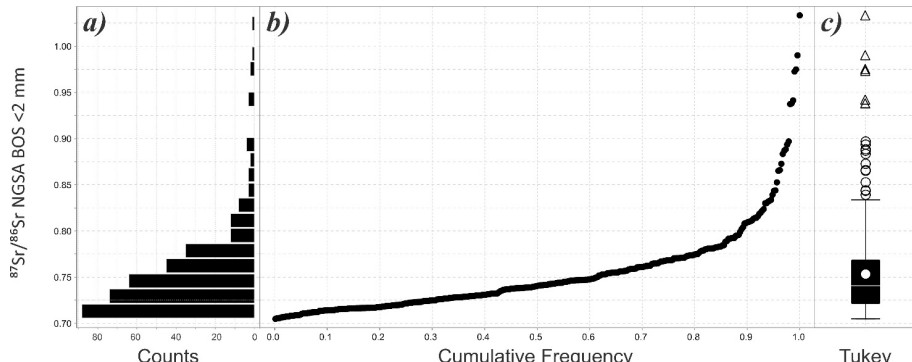

**Figure 3. Univariate distribution of the $^{87}Sr/^{86}Sr$ data ($n$ = 357) from the northern Australia Sr isotope study area: (a) histogram (20 bins 0.016 wide); (b) cumulative frequency plot; and (c) Tukey boxplot (mean = white dot; outliers = circles; and far outliers = triangles) (Tukey, 1977). Sample medium is the < 2 mm fraction of NGSA Bottom Outlet Sediment (BOS) or equivalent (see text for further information).**


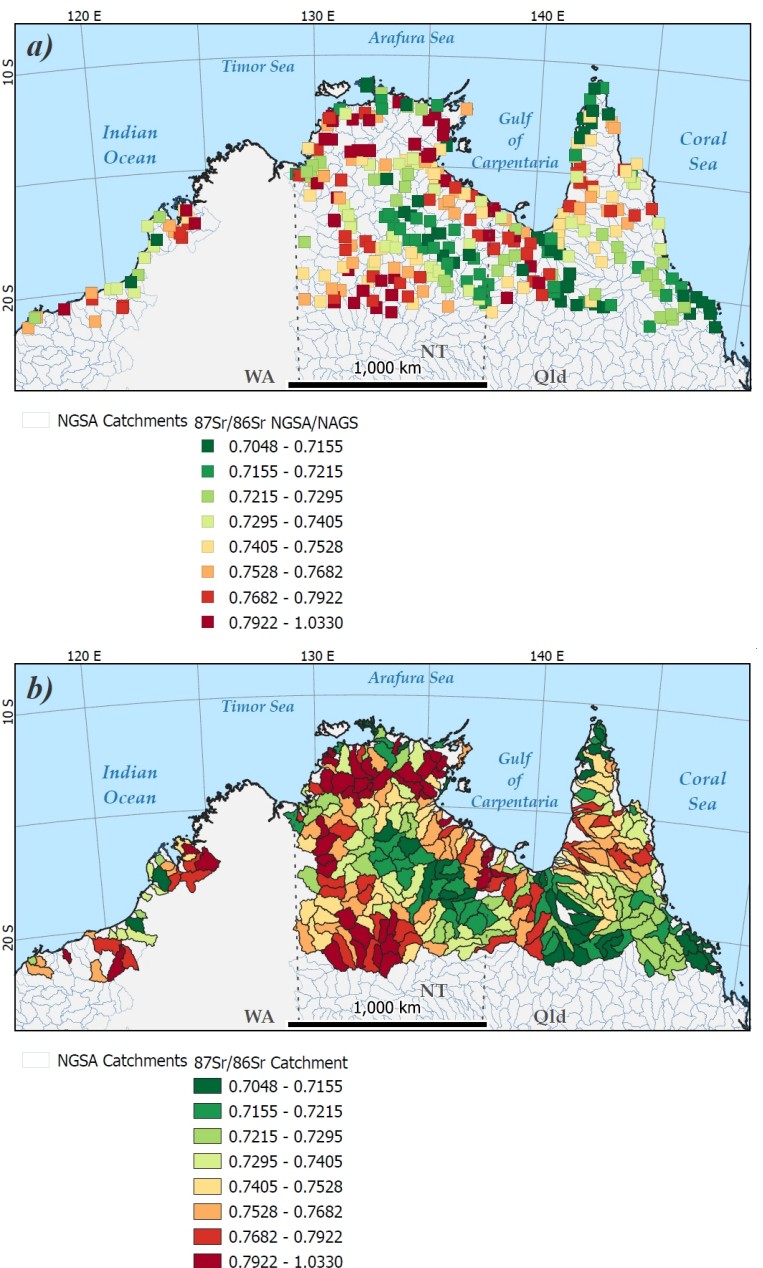

**Figure 4. Strontium isoscape for the northern Australia study area with (a) data points classed by quantiles at the sampling sites; and (b) as whole catchments coloured by same colour ramp. Sample medium is the < 2 mm fraction of NGSA Bottom Outlet Sediment (BOS) or equivalent (see text for further information). Map projection: Albers equal area.**




# 5 Discussion

## 5.1 Comparison with other datasets

A comparison of the present results with selected soil $^{87}$Sr/$^{86}$Sr datasets from around the world is offered in Table 1. All discoverable data from Australia and the southern hemisphere comprising more than a handful of sites were included, whilst
only selected northern hemisphere datasets focusing on recent and large-scale datasets were included. The table combines both bioavailable/exchangeable and bulk/total $^{87}$Sr/$^{86}$Sr data, representing variable soil grain-size fractions (where reported), parent materials and landuses. Despite that variability, one can make several observations. Firstly, the northern Australia Sr isotope dataset has the highest maximum (1.0330) and largest range (0.3282) of $^{87}$Sr/$^{86}$Sr values amongst those compiled. Secondly, there is no observed standard protocol for collecting or preparing soils for either bioavailable or total $^{87}$Sr/$^{86}$Sr
determination. Thirdly, total $^{87}$Sr/$^{86}$Sr datasets tend to have a wider range and greater variance than bioavailable ones, at least partly accounting for the higher values encountered here.



**Table 1.** Comparison of $^{87}Sr/^{86}Sr$ data from this study with selected datasets from around the world, with emphasis on Australia and the southern hemisphere (count, minimum, median, mean, standard deviation, maximum, and range). Regions = Aus: Australia; Qld: Queensland; SA: South Australia; SE: Southeastern; Vic: Victoria; WA: Western Australia. Digestions = AcAc: Acetic Acid ($CH_3COOH$); AmAc: Ammonium Acetate ($CH_3COONH_4$); AmNt: Ammonium Nitrate ($NH_4NO_3$); AR: Aqua Regia.

| Region | Sample Type | Digestion | $n$ | Min | Med | Mean | SD | Max | Range | Source |
|---|---|---|---|---|---|---|---|---|---|---|
| *S Hemisphere: Australia* | | | | | | | | | | |
| Northern Aus | Soil (on alluvium), < 2 mm | Milled, HF + $HNO_3$ + AR | 357 | 0.7048 | 0.7390 | 0.7527 | 0.0484 | 1.0330 | 0.3282 | This study |
| SE Aus | Soil (on alluvium), < 2 mm | Milled, HF + $HNO_3$ + AR | 112 | 0.7089 | 0.7199 | 0.7220 | 0.0736 | 0.7511 | 0.0422 | de Caritat et al., 2022b |
| Mostly SA, Vic; 19 sites | Soil (calcrete) | AcAc | 26 | 0.7061 | 0.7102 | 0.7115 | 0.0051 | 0.7329 | 0.0267 | Quade et al., 1995 |
| SE Aus; Murray Darling Basin | Alluvium, < 2 µm | HF + $HNO_3$ | 26 | 0.7080 | 0.7163 | 0.7267 | 0.0188 | 0.7751 | 0.0672 | Gingele & De Deckker, 2005 |
| Southern Aus; 8 sites | Soil (calcrete) | AcAc | 46 | 0.7094 | 0.7151 | 0.7168 | 0.0058 | 0.7374 | 0.0280 | Dart et al., 2007 |
| | Soil (calcrete) | $HNO_3$ + HF + HCl | 38 | 0.7125 | 0.7382 | 0.7538 | 0.0588 | 0.9985 | 0.2860 | Dart et al., 2007 |
| Canning Coast, WA | Dust, < 2 µm | AcAc | 14 | 0.7141 | 0.7361 | 0.7385 | 0.0163 | 0.7727 | 0.0586 | De Deckker, 2019 |
| Cape York Peninsula, Qld | Soil, < 2 mm | AcAc | 93 | 0.7075 | 0.7227 | 0.7252 | 0.0154 | 0.7911 | 0.0835 | Adams et al., 2019 |
| *S Hemisphere: Other* | | | | | | | | | | |
| Brazil, 3 regions | Soil (agricultural), < 2 mm | AmAc | 3 | 0.7122 | 0.7132 | 0.7129 | 0.0006 | 0.7133 | 0.0011 | de Almeida, 2021 |
| Southern Chile | Soil (forest) | AmAc | 22 | 0.7091 | 0.7098 | 0.7101 | 0.0007 | 0.7119 | 0.0029 | Kennedy et al., 2002 |
| Southern Peru | Soil (agricultural) | Ashed, AmAc | 114 | 0.7020 | 0.7076 | 0.7077 | 0.0017 | 0.7189 | 0.0169 | Knudson et al., 2014 |
| South Africa, 4 regions | Soil (agricultural), < 180 µm | $HNO_3$ + $H_2O_2$ | 67 | 0.7081 | 0.7130 | 0.7126 | 0.0020 | 0.7159 | 0.0078 | Vorster at al., 2010 |
| Southern New Zealand | Soil, < 2 mm | AmNt | 83 | 0.7040 | 0.7079 | 0.7078 | 0.0013 | 0.7112 | 0.0072 | A. Martin, pers. comm., 2021 |

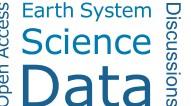

| Region | Sample Type | Digestion | $n$ | Min | Med | Mean | SD | Max | Range | Source |
|---|---|---|---|---|---|---|---|---|---|---|
| *N Hemisphere* | | | | | | | | | | |
| Britain | Soil | $H_2O$ | 26 | 0.7074 | 0.7093 | 0.7092 | 0.0010 | 0.7115 | 0.0041 | Evans et al., 2010 |
| France | Soil, < 2 mm | AmNt | 524 | 0.7033 | 0.7113 | 0.7119 | 0.0042 | 0.7308 | 0.0275 | Willmes et al., 2014 |
| Europe | Soil (grazing), < 2 mm | AmNt | 565 | 0.7038 | 0.7090 | 0.7103 | 0.0052 | 0.7523 | 0.0485 | Hoogewerff et al., 2019 |
| | Soil (agricultural), < 2 mm | AmNt | 622 | 0.7038 | 0.7093 | 0.7112 | 0.0063 | 0.7596 | 0.0558 | Hoogewerff et al., 2019 |
| Israel | Soil, < 2 mm | AmNt | 60 | 0.7058 | 0.7086 | 0.7085 | 0.0008 | 0.7102 | 0.0044 | Moffat et al., 2020 |
| Cambodia | Soil | Ashed, HF | 60 | 0.7034 | 0.7115 | 0.7130 | 0.0060 | 0.7424 | 0.0389 | Shewan et al., 2020 |
| | Soil | Ashed, AmNt | 46 | 0.7037 | 0.7110 | 0.7117 | 0.0039 | 0.7204 | 0.0167 | Shewan et al., 2020 |
| Japan | Soil, < 2 mm | $H_2O_2$ | 13 | 0.7067 | 0.7085 | 0.7088 | 0.0016 | 0.7129 | 0.0062 | Oishi, 2021 |
| Italy | Soil | Exchangeable and bulk | 273 | 0.7053 | 0.7091 | 0.7099 | 0.0027 | 0.7238 | 0.0185 | Lugli et al., 2022 |

## 5.2 Top-bottom relationship

Based on the 15 sites where both TOS and BOS samples were analysed, a strong correlation (Figure 5) between those two depths is found:

$$(^{87}Sr/^{86}Sr)_{BOS} = 0.9913 \times (^{87}Sr/^{86}Sr)_{TOS} + 0.0069 \ (r^2 = 0.97; p < 0.001; n = 15). \tag{1}$$

Equation 1 was used to infer BOS $^{87}Sr/^{86}Sr$ from those sites where only TOS samples could be obtained (i.e. 3 NGSA and 28 NAGS sites), effectively deriving a 'BOS-equivalent' $^{87}Sr/^{86}Sr$ value for the purposes of performing internally consistent statistical and spatial analysis.

As a result of the robust correlation between TOS and BOS, the BOS (or subsurface) Sr isoscapes of Figure 4 can be recalculated to TOS (or surface) Sr isoscapes by replacing the legend values for the eight quantile class boundaries by the values 0.7050 - 0.7156 - 0.7215 - 0.7293 - 0.7401 - 0.7522 - 0.7674 - 0.7910 - 1.0276.





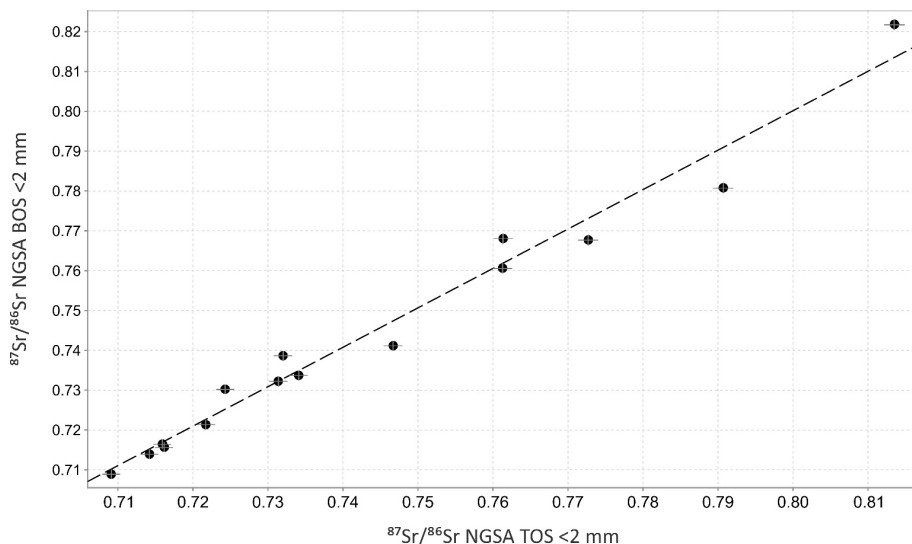


**Figure 5. Scatterplot of the BOS (deep) versus TOS (surface) $^{87}$Sr/$^{86}$Sr data ($n$ = 15) from the northern Australia Sr isotope study area with total uncertainty derived from field duplicates (0.17 %; grey plusses) and linear square regression line (dashed line).**

## 5.3 Relationship to bedrock

When grouped by age of their respective geological region, the new $^{87}$Sr/$^{86}$Sr data show a general trend of increasing

$^{87}$Sr/$^{86}$Sr with increasing bedrock age, as illustrated by the box and scatter plots of Figure 6. In this figure, average numerical ages (mid-points between beginning and end ages) are attributed to each geological period group. The observed trend is consistent with one of the known controls on mineral (and rock) $^{87}$Sr/$^{86}$Sr values, namely their age.

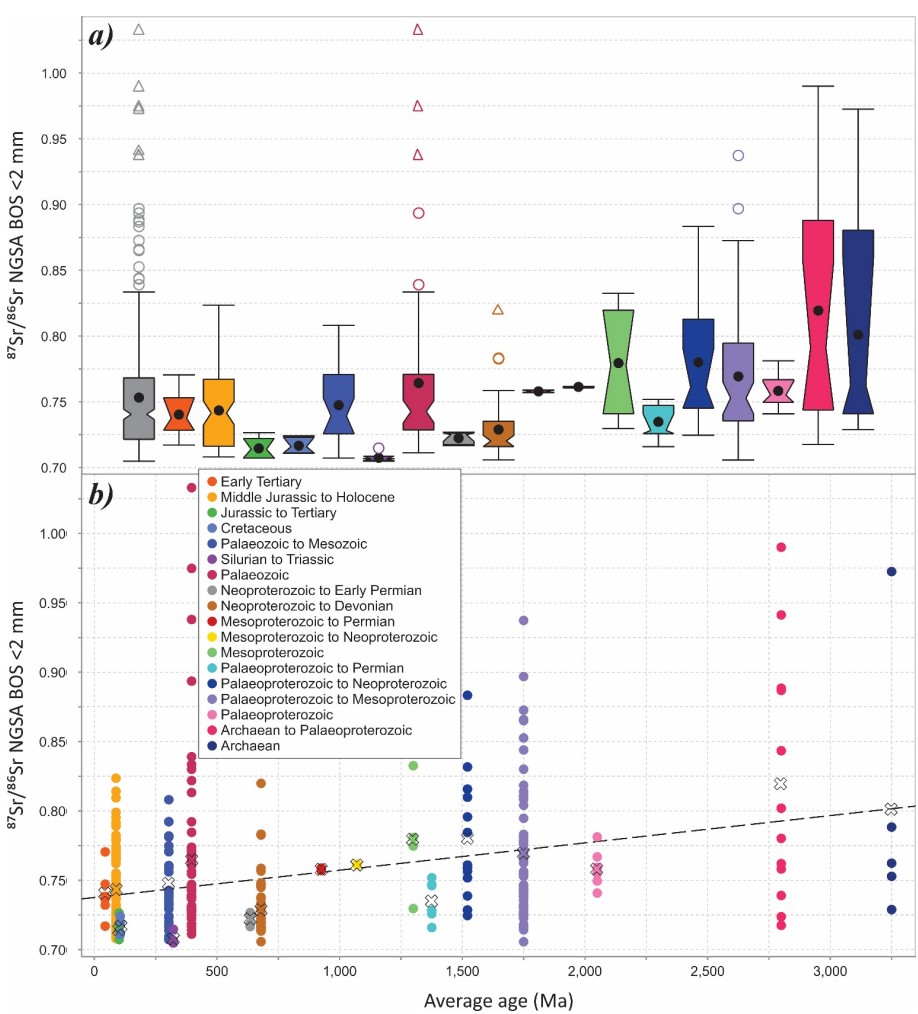

**Figure 6. Distribution of the $^{87}$Sr/$^{86}$Sr data from the northern Australia Sr isotope study area by geological region age: (a) notched Tukey boxplots (Tukey, 1977) of, left to right, the whole dataset first (grey; *n* = 357), then sorted by average age (Early Tertiary, *n* = 6; Middle Jurassic to Holocene, *n* = 80; Jurassic to Tertiary, *n* = 5; Cretaceous, *n* = 4; Palaeozoic to Mesozoic, *n* = 28; Silurian to Triassic, *n* = 6; Palaeozoic, *n* = 61; Neoproterozoic to Early Permian, *n* = 7; Neoproterozoic to Devonian, *n* = 49; Mesoproterozoic to Permian, *n* = 2; Mesoproterozoic to Neoproterozoic, *n* = 2; Mesoproterozoic, *n* = 4; Palaeoproterozoic to Permian, *n* = 7; Palaeoproterozoic to Neoproterozoic, *n* = 13; Palaeoproterozoic to Mesoproterozoic, *n* = 59; Palaeoproterozoic, *n* = 7; Archaean to Palaeoproterozoic, *n* = 12; and Archaean, *n* = 5); and (b) scatterplot against average age of the geological region with group averages (cross) and linear square regression line (dashed).**

Another known control on Sr isotopic values is the Rb content, which over time contributes $^{87}$Sr by radiogenic decay. Closer examination of the geology in the catchments with the 10 highest $^{87}$Sr/$^{86}$Sr values indicates that they contain lithologies likely to be enriched in Rb relative to Sr (e.g. felsic, felsic-to-intermediate igneous rocks, medium- to high-grade metamorphic rocks) as shown by detailed maps in the Supplement. Of the five top-ten $^{87}$Sr/$^{86}$Sr values in the northern NT, three are from the Pine Creek geological region. The two northernmost of these (samples #2007190099 and #2007191390) with $^{87}$Sr/$^{86}$Sr = 0.9413 and 0.9900, respectively, are in catchments downstream of 1780-1790 Ma I-type granite intrusions



that have been reported to have $^{87}Sr/^{86}Sr$ from 0.71 to 8.13, $^{87}Rb/^{86}Sr$ = ~1 to 290, up to 956 mg/kg Rb and up to 459 mg/kg Sr (Riley, 1980, table1).

The elevated samples from the WA coast are also generally proximal to, or downstream of, felsic igneous lithologies (including the granitoid and gneiss underlying the Mallina Basin; Van Kranendonk et al., 2002) and are consistent with $^{87}Sr/^{86}Sr$ values reported on clay fraction sediments from the region by De Deckker (2019) as shown in the Supplement. The northernmost samples along the coast of WA overlap with the King Leopold Orogen abutting the western flank of the Kimberley geological region. Few $^{87}Sr/^{86}Sr$ data are available for this region, but those that exist within the Hooper Complex
(Griffin et al., 2000) indicate radiogenic $^{87}Sr/^{86}Sr$ values: 0.7708-0.9620 for the Whitewater Volcanics, 0.7336-0.8524 for the Kongorow granite, and 0.7353-0.7602 for the Lennard granite (Bennett and Gellatly, 1970, table 2). Thus, observed elevated $^{87}Sr/^{86}Sr$ values of 0.8080, 0.7709, and 0.7923 in three sediment samples (#200719103, #2007190317, and #2007190966, respectively) near Derby, WA, are entirely compatible with these bedrock source rocks.

Page et al. (1980, table 5) published Rb and Sr data from the Alligator Rivers uranium field rocks, including from the
Nimbuwah complex granitoid close to Cooper Creek, < 20 km north of the Nabarlek uranium deposit (their sample #7212.4063). For this sample they reported $^{87}Rb/^{86}Sr$ = 1.25-1.47, $^{87}Sr/^{86}Sr$ = 0.7387-0.7442, 146-164 mg/kg Rb and 320-336 mg/kg Sr. Our sample (#2007190710) from this same catchment has a $^{87}Sr/^{86}Sr$ value of 0.8019 and is located on a felsic granite polygon.

Black et al. (1983, tables 3 and 5) reported calculated initial rock $^{87}Sr/^{86}Sr$ ratios in the range 0.76 to 0.92 from granites,
including the Wangala and the Haverson granites, in the northern Arunta geological region (southern NT). These intrusions intersect three sampled NGSA catchments for which we obtained elevated $^{87}Sr/^{86}Sr$ values, namely 0.8140, 0.7763, and 0.8651 (samples #2007190727, #2007190562, and #2007190123, respectively). These catchments are in turn just upstream of some of our top-ten $^{87}Sr/^{86}Sr$ values (between 0.8935 and 1.0330).

In the Georgetown geological region of northwestern Qld, broad consistency between whole rock $^{87}Sr/^{86}Sr$ and catchment
sediment $^{87}Sr/^{86}Sr$ is also observed. For instance, our sample #2007190858, from a catchment that drains mostly the Esmeralda granite (average whole rock initial $^{87}Sr/^{86}Sr$ of 0.7314 ± 0.0116; Black, 1973, table 2) on the western edge of the Georgetown Inlier, has a relatively elevated $^{87}Sr/^{86}Sr$ value of 0.7467. Conversely, the three catchments that mostly directly drain the less radiogenic Newcastle Range Volcanics in the centre of the Inlier, which have a lower average whole rock initial $^{87}Sr/^{86}Sr$ of 0.7152 ± 0.0011 (Black, 1973, table 2), have correspondingly lower sediment $^{87}Sr/^{86}Sr$ values of 0.7318
(#2007190572), 0.7268 (#2007190348), and a field duplicate pair of 0.7258 (#2007190548) and 0.7282 (#2007190548), respectively.

### 5.4 Relationship to mineralisation

Figure 7 illustrates the range of $^{87}Sr/^{86}Sr$ values of catchments that contain known mineralisation of various types. Forty four NGSA catchments host 97 mineral occurrences as catalogued by GA (2022a). These resources include 19 'Base Metal-Zn-
Pb' – Zn, Pb (Cu, Ag); 17 'Battery/Alloy Metals' – Ni, Co, Mn, V, Mo, Mg; 13 'Precious Metals' – Au, Ag; 12 'Fertilizer



Elements' – P, K; 6 'Rare Earth Elements'; 6 'Other Metals' – Sn, Sb, W, Ta, Nb; 5 'Iron Ore' – Fe; and 5 'Base Metal-Cu' – Cu (Zn, Pb, Ag, Au) occurrences. Whilst the vast majority of mineral occurrences are found in catchments with a sediment outlet $^{87}Sr/^{86}Sr$ signature << 0.84, six outlier occurrences are associated with higher catchment outlet sediment $^{87}Sr/^{86}Sr$ values: 0.8434 (Coronation Hill Pt, Pd deposit in the catchment containing sample #2007191552); 0.9373 (Nolans Bore hard

rock REE and Mount Peake vanadium deposits the catchment containing sample #2007191112); 0.9748 (Batman Au deposit in the catchment containing sample #2007191329); and highest of all 0.9900 (Browns Pb-Zn and the Browns Co deposits in the catchment containing sample #2007191387). Some mineral occurrence types are not found in catchments with $^{87}Sr/^{86}Sr$ signature less radiogenic than 0.75 in this study: the minima reported for commodity groups 'Platinum Group Elements' being 0.7529; 'Rare Earth Element'/'Heavy Mineral Sands' 0.7606; and 'Diamond' 0.7567. Whilst it is unlikely that mineral

deposits themselves impart a significant control on the $^{87}Sr/^{86}Sr$ of sediment collected down-catchment given their usually limited size, the hosting country rock, however, may potentially be more widely affected by mineral system processes, and appears to record some Sr isotopic effect that may be useful for mineral vectoring upon further investigation. One of the most radiogenic $^{87}Sr/^{86}Sr$ signatures (0.9373) is found for the catchment hosting both the Mount Peake V deposit (Simandl and Paradis, 2022) and the Nolans Bore hydrothermal REEs deposit (Schoneveld et al., 2015). Hydrothermal or magmatic

REEs deposits are not usually characterised by highly radiogenic Sr signatures (e.g. 0.7029-0.7262 at Bayan Obo, Le Bas et al., 1997; 0.701-0.708 in global review of carbonatite deposits, Bolonin, 2019; 0.7037-0.7062 at Caotan, Wei et al., 2020) and nor does Nolans Bore specifically (0.7054 to 0.7079; Huston et al., 2016). Thus, the elevated $^{87}Sr/^{86}Sr$ reported here from this catchment is attributed to the radiogenic bedrock occurring in the study area and perhaps in upstream catchments immediately to the south of it.

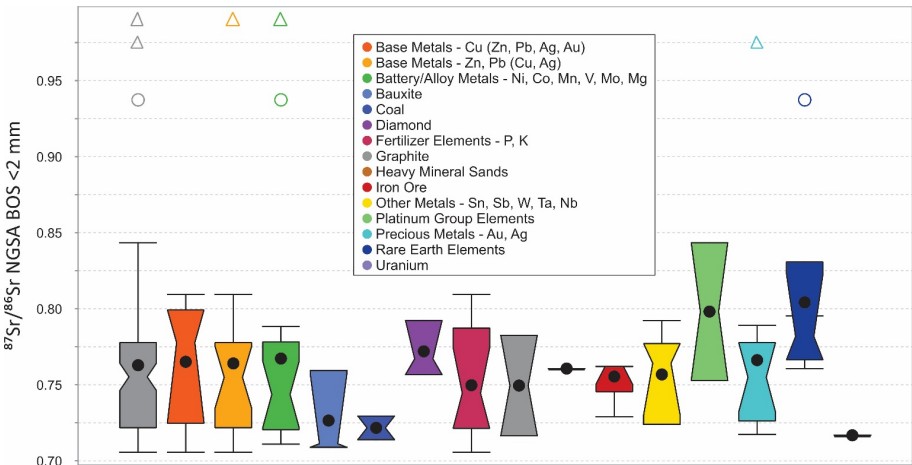


**Figure 7. Distribution of the $^{87}Sr/^{86}Sr$ data from the northern Australia Sr isotope study area by mineral occurrence hosted in same catchment. From left to right, All ($n$ = 97); 'Base Metals – Cu (Zn, Pb, Ag, Au)' (5); 'Base Metals – Zn, Pb (Cu, Ag)' (19); 'Battery/Alloy Metals – Ni, Co, Mn, V, Mo, Mg' (17); 'Bauxite' (3); 'Coal' (2); 'Diamond' (3); 'Fertilizer Elements – P, K' (12); 'Graphite' (2); 'Heavy Mineral Sands' (1); 'Iron Ore' (5); 'Other Metals – Sn, Sb, W, Ta, Nb' (6); P'latinum Group Elements' (2);**

**'Precious Metals – Au, Ag' (13); 'Rare Earth Elements' (6); and 'Uranium' (1).**



## 6 Data availability

The new spatial Sr isotope dataset for the northern Australia region is publicly available from https://dx.doi.org/10.26186/147473 (de Caritat et al., 2022a) and through the Geoscience Australia portal https://portal.ga.gov.au/restore/cd686f2d-c87b-41b8-8c4b-ca8af531ae7e (last access: 15 December 2022). Metadata are also available through the Geoscience Australia portal (https://portal.ga.gov.au/metadata/geochronology-and-isotopes/isotopes/rbsr-isotope-points/4cacd9e8-3340-4c27-99fe-48d404e67ca8, last access 15 December 2022) (GA, 2022b).

## 7 Conclusions

Three hundred and seventy two new strontium (Sr) isotopic compositions ($^{87}$Sr/$^{86}$Sr) are reported from 357 catchment outlet sediment samples from northern Australia (north of 21.5 °S). The analysed material originates from the sample archives of the National Geochemical Survey of Australia (NGSA; $n$ = 344) and Northern Australia Geochemical Survey (NAGS; 28) projects, both of which targeted overbank or floodplain landforms near the outlet of large catchments. The sampled catchments together cover 1.536 million km$^2$ of northern Australia. For the most part, bottom outlet sediment (BOS) samples, retrieved mostly by augering to, on average, 0.6 to 0.8m depth, were analysed. A few top outlet sediment (TOS) samples, collected from the top 0.1 m of soil, however, were also included, notably all the NAGS samples and 18 NGSA samples. Total digestion of milled < 2 mm grain-size fractions from these sediments yielded a wide range of $^{87}$Sr/$^{86}$Sr values from a minimum of 0. 7048 to a maximum of 1.0330.

The present study represents the largest Sr isoscape in the southern hemisphere to-date, a region that is critically under-represented in terms of isotopic data worldwide. We found a very strong correlation between the $^{87}$Sr/$^{86}$Sr values in the BOS and TOS samples, allowing us to confidently infer 'BOS-equivalent' values where only TOS samples were analysed. A map of the $^{87}$Sr/$^{86}$Sr distribution (isoscape) across northern Australia reveals spatial patterns reflecting the ages and lithologies of the source material for the sediment, which is principally carried down catchment by fluvial processes. There is an overall increase in $^{87}$Sr/$^{86}$Sr observed with increasing age of the geological region intersected at the sampling points. In areas of outcropping or subcropping felsic igneous rocks, relatively radiogenic Sr signatures are observed, and conversely for areas of mafic igneous rocks or marine sediments. The new Sr isotopic data are also interrogated in terms of the mineral occurrences found in their catchment. Whilst most mineral occurrences in the region are found in catchment with an $^{87}$Sr/$^{86}$Sr signature << 0.84, six outlier occurrences are associated with higher values. Some mineral occurrence types ('Platinum Group Elements', 'Rare Earth Element'/'Heavy Mineral Sands', and 'Diamond') are not found in catchments with $^{87}$Sr/$^{86}$Sr signature less radiogenic than 0.75 in this study. Whether mineral system or regional petrogenetic processes are responsible for the association of relatively elevated $^{87}$Sr/$^{86}$Sr with mineral deposits is a subject for further investigation.

Although we have focused the discussion of the new $^{87}$Sr/$^{86}$Sr data on sediment sources in terms of rock ages and types, potential applications of the present isoscape could be extended to studies of mineralization, hydrology, food tracing, dust provenancing/sourcing, and historic migrations of people and animals.

**Supplement.** The supplement related to this article is available online at: https://doi.org/10.5194/essd-xx-xxxx-xxxx-
supplement.

**Author contributions.** PdC provided the concept, samples, funding, data curation, analysis and visualization, and manuscript writing and editing. AD provided technical guidance, resources and supervision, data curation, and manuscript editing. FD provided technical support and data curation.


**Competing interests.** The contact author has declared that none of the authors has any competing interests.

**Disclaimer.** PdC publishes with the permission from the Chief Executive Officer, Geoscience Australia.

Publisher's note: Copernicus Publications remains neutral with regard to jurisdictional claims in published maps and institutional affiliations.

**Acknowledgments** The National Geochemical Survey of Australia (NGSA) project would not have been possible without Commonwealth funding through the Onshore Energy Security Program (http://www.ga.gov.au/ngsa, last access: 15
December 2022), and Geoscience Australia appropriation. The Northern Australia Geochemical Survey (NAGS) project was funded under the Exploring for the Future Program (https://eftf-production.ga.gov.au/northern-australia-geochemical-survey, last access: 15 December 2022), and Geoscience Australia appropriation. Collaboration with the geoscience agencies of all states and the Northern Territory is gratefully recognised. We acknowledge all land owners and custodians, whether private, corporate, and/or traditional, for granting access to the field sites for the purposes of sampling. We thank Geoscience
Australia laboratory staff for assistance with preparing the samples. The Sr isotopic analyses reported here were funded by the Exploring for the Future (EFTF 2020–2024) Program (https://www.eftf.ga.gov.au/, last access: 15 December 2022) funded by the Australian Government.

**Financial support.** This research has been supported by the Australian Government's Exploring for the Future program
(https://www.eftf.ga.gov.au/about, last access: 15 December 2022).

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
