# Peer review of "A strontium isoscape of northern Australia"

_Earth System Science Data, 2022_

## Author Comment (AC1)

Reply to Comment on essd-2022-446 by Ian Moffat (Referee)

Referee comment on "A strontium isoscape of northern Australia" by Patrice de Caritat et al., Earth Syst. Sci. Data Discuss., https://doi.org/10.5194/essd-2022-446-RC1, 2023

**General Comments:**

Thank you to the authors for this important data set, which addresses an expansive deficiency in strontium isotope mapping in Northern Australia. The methodology of this study is robust, and it will provide an important tool to facilitate future research in this area.

Several aspects of this study are of particular note. The analysis of the relationship between the "top outlet sediment" and "bottom outlet sediment" is novel and expands the interpretability of this data set. The consideration of strontium isotope values compared to mineralisation is also a very interesting inclusion.

Thank you for the comments.

**Specific Comments:**

I suggest the authors make explicit that food tracing, provenance and archaeological applications of strontium isotope tracing (referred to specifically in lines 64-68) rely on bioavailable rather than "whole rock" values and so don't benefit directly from this data set.

The sentence now clarifies that these studies use bioavailable Sr.

The decision to analyse catchment outlet sediments is understandable given the availability of these samples from the National Geochemical Survey of Australia, however this methodology differs from most other strontium isoscape studies and has important implications for interpretation of these results. I suggest the authors elaborate on these issues, perhaps as a new section in the discussion.

This is an important point and we have strengthened our discussion of it in Subsection "5.1 Comparison with other datasets", which we find most appropriate location for this. The following sentences have been added:

*These issues complicate data compilation and integration across projects/countries, but do not preclude them. Indeed, through contemporaneous data analytics, including artificial intelligence and machine learning, it is likely that the relationships between bioavailable and total Sr isotope values and a host of other environmental variables (including from climatic, topographic, biotic and geoscientific categories) can be teased out and high-spatial resolution models/predictions of bioavailable or total Sr isotope distribution can be derived (e.g. Bataille et al., 2020). This is indeed a future research direction we propose/support for isoscape science in general, and an Australian Sr isoscape in particular.*

The data set presented by the authors in this paper is extremely useful and interesting however it could be considerably enhanced by including the bedrock geology, age and stratigraphic unit (as summarised in Figure 6), notwithstanding that these samples have been collected from fluvial systems.

Thank you for the suggestion. We present a new Sr isotopic dataset here, and do not pretend to provide a complete dataset of all possible variables all potential users may need or wish to have. We leave this degree of data manipulation and integration to future users as they are bespoke customisations we cannot start to predict. The important and relevant contribution we make here,

in our view, is to put a significant new dataset on the table, so we deliberately want to remain focussed on this in this paper.

**Technical Corrections:**

Figure 1 is extremely small in the manuscript and the sample locations use the same colour as the Towns and Places, can I suggest this is revised to make it easier to view?

As this point is also raised by another reviewer, we suggest the ==Editor== requests the production team prints this figure in a larger format than currently shown. We have prepared a new Figure 1 with Towns and Places more clearly distinguishable (changed from gray to orange).

---

## Author Comment (AC2)

Reply to Comment on essd-2022-446 by Malte Willmes (Referee)

Referee comment on "A strontium isoscape of northern Australia" by Patrice de Caritat et al., Earth Syst. Sci. Data Discuss., https://doi.org/10.5194/essd-2022-446-RC2, 2023

**General comments**

The manuscript "A strontium isoscape of northern Australia" by Caritat et al. presents a new large-scale 87Sr/86Sr isotope dataset for Northern Australia based on archival fluvial sediment samples. The scope of the study is very impressive, and the data are going to be very useful for the science community. The analytical methodology and statistical evaluation of the data are appropriate and provide confidence in the provided results. Specifically, the use of archived sediment samples that should represent a mixture of the varied geology is very interesting and overcomes the lack of samples from Northern Australia. Overall, I believe this is a very important contribution and it is great to see that the authors have chosen to make this data available.

Thank you for the comments.

**Specific comments**

I think it would benefit the reader to make it very clear that the data presented here are total Sr and not bioavailable Sr. This does get mentioned in the manuscript, but I think it should be front and center to avoid any potential confusion and incorrect use of the data. I am also interested in why the bulk Sr extraction was chosen rather than bioavailable? I think elaborating on this in the discussion would be useful and interesting. Is there any use in getting bioavailable Sr out of these samples?

The fact that we use total Sr, which was already made several times, has been emphasised in the text (e.g. italicised *total/bioavailable* in Abstract, Introduction and Conclusions). The reason we chose this method is because our focus is on geological processes rather than bio-mediated processes. The following sentence was added at the end of the Introduction:

*The choice of total rather than bioavailable Sr as the focus of this work was driven by an emphasis on geological sources and processes.*

In the quality assessment section (Line 222-228): To confirm the data are interpreted based on the 3rd decimal place because the within sample variability was high? Why is that so? Is that a result of the total Sr extraction method, the type of these sediments, or another process? This should be elaborated on in the discussion. For example, would you recommend analyzing more sediment samples to get a better handle on this? Or a different extraction method? Or a different sample type?

We clarified this by modifying the sentence in Subsection "3.3 Quality Assessment":

*This relatively low precision obtained for field duplicates is attributed to heterogeneity of the alluvial deposits*

Analysing more samples will improve results, but there has to be a balance between extent of coverage and density of sampling because resources are always finite. In the NGSA the choice was made to favour the former and we have to live with this when working with these archive samples. Perhaps the only thing that could be made differently in a future survey is to collect field duplicates much closer than 80 m apart, perhaps within 10 m of each other. So in many ways the NGSA sampling represents the worst case scenario in terms of representing the homogeneity of the target landform.

**Technical corrections**

Abstract

I focused on the abstract as this is the most widely read part of the paper. The following comments are all suggestions that I hope improve clarity.

Line 8: Maybe add "useful as a tracer" or something similar. Just useful seems not very informative.

Done.

Line 10: I think it should be "archived".

Done.

Line 11: Suggest removing the reference style (last accessed) from the abstract (here and at the end).

Removed.

Line 12-14: This is a lot of method detail for an abstract. Since the paper is open access and the method section will be available to all to read I suggest to condense this to a single sentence. "Total Sr was extracted and measured…"

We understand the viewpoint, however, as our methods are dissimilar to those employed by e.g. anthropologists (as emphasised by this and other Reviewers), we prefer to leave this amount of detail in the Abstract. If deemed excessively long (we don't think so), we're happy to cut this down somewhat at final stage.

Line 16: Why preliminary?

Removed.

Line 21: Can you add a range to the carbonate units. Also, the word "signature" at least to my ears implies some type of unique value so maybe the word "range" or "ratio" is better here.

Done, 'signature' changed to 'values' and medians for carbonate and mafic/ultramafic rocks added. A corresponding paragraph has been added in the Discussion (Subsection 5.3):

*The floodplain sediment Sr isotopic values recorded in areas dominated by sedimentary carbonate and mafic/ultramafic igneous rocks are usually within an intermediate range between the more radiogenic and unradiogenic end-members discussed above. Indeed, samples sited within 0.1 degree (~10 km) of lithologies recorded as sedimentary carbonate and mafic/ultramafic igneous rocks have median values of 0.7387 (n = 96) and 0.7422 (n = 42), respectively.*

Line 27: How should these data be used in archaeological, forensic, and ecological studies? The sentence from Line 76 might actually be a good fit here. "The present study affords an opportunity to further redress this deficiency and will reduce the northern hemisphere bias in future global 87Sr/86Sr models."

We don't think this needs to be spelt out in the Abstract any more than it is, as the words *'future models derived therefrom'* are sufficiently explicit. The details are now covered by the added sentence in Subsection 5.21:

*Indeed, through contemporaneous data analytics, including artificial intelligence and machine learning, it is likely that the relationships between bioavailable and total Sr isotope values and a host*

*of other environmental variables (including from climatic, topographic, biotic and geoscientific categories) can be teased out and high-spatial resolution models/predictions of bioavailable or total Sr isotope distribution can be derived (e.g. Bataille et al., 2020).*

Introduction

Line 40-70: I don't believe all these references are needed or useful for the reader. Maybe focusing on 2-3 key references per statement would improve this intro.

As recent reviews of Sr isotope work in the geosciences are rare, we find this short review (~1 page) has merit, and since other Reviewers have not raised an issue with it, we prefer to leave it in.

Figure 1: Great figure but small in the pdf and symbology is hard to distinguish (maybe different symbols could improve this). Actually all figures could be larger (full width).

We have redone Figure 1 for clarity, as Reviewer 1 also raised that point. We leave it to the ==Editor== and the production team to ensure Figures are of an appropriate size.

Data analysis

Line 230: Why choose Excel rather than R or python for the data analysis? One of the main benefits of using a scripting language would be that others could reproduce the results (and figures) directly.

That was the tool available to us at the time. The raw data are released so future researchers can manipulate and record their analysis with R or python if they wish.

Figure 4: Remove one of the legends to save space and as before make the figure larger

The legends refer to points and catchments, respectively. We prefer to leave them as is, and leave it to the ==Editor== and the production team to ensure Figures are of an appropriate size.

Table 1: Really useful compilation! Thank you.

Thank you for the comments.

Conclusions

Line 373-381: This could be removed to provide more focus in the conclusion. The number of samples and the range could be incorporated into the following sentences.

We feel that many readers may only read the Abstract and then perhaps the Conclusions, and so believe that it is important that the main parameters of the research are repeated in the latter.

Also again be very specific about how these data could be applied in provenance studies in archaeology/forensics.

The words *'and modelling derived therefrom'* have been added.

---

## Author Comment (AC3)

Reply to Comment on essd-2022-446 by Jodie Pritchard (Referee)

Referee comment on "A strontium isoscape of northern Australia" by Patrice de Caritat et al., Earth Syst. Sci. Data Discuss., https://doi.org/10.5194/essd-2022-446-RC3, 2023

**General comments:**

Congratulations to the authors. This paper contains a great quality, highly valuable largescale bulk soil 87/86Sr dataset that will be applicable to many areas of scientific research, for an area of Australia where there was previously very little information available. Within the paper the bulk soil 87/86Sr dataset is compared with catchment specific information including bedrock age, lithology and mineralisation.

Thank you for the comments.

**Specific comments:**

Soil samples were taken at the bottom of the catchment which is a unique and pragmatic approach for maximising coverage. How does this compare with previous approaches and effect interpretation?

This choice is a legacy of using a sample archive collected for the purposes of geochemical mapping. In large-scale geochemical mapping, sampling overbank or floodplain sediments is a well-established and widely used technique (references below). Several publications and our experience with the NGSA show that the floodplain sediment technique provides a realistic representation of the *average* conditions in the catchment. Having said that, there is no question that additional subsampling within catchment would add valuable granularity to the maps. But this is a balance between extent of coverage and density of sampling because resources are always finite. We believe that interpretations at this sub-continental scale are appropriate based on this sampling density.

BØLVIKEN, B., STOKKE, P.R., FEDER, J. & JOSSANG, T., 1992. The fractal nature of geochemical landscapes. Journal of Geochemical Exploration, 43: 91-109.

OTTESEN, R.T., BOGEN, J., BØLVIKEN, B. & VOLDEN, T., 1989. Overbank sediment: a representative sample medium for regional geochemical sampling. Journal of Geochemical Exploration, 32: 257-277. DOI: 10.1016/0375-6742%2889%2990061-7.

DARNLEY, A.G., BJÖRKLUND, A., BØLVIKEN, B., GUSTAVSSON, N., KOVAL, P.V., PLANT, J.A., STEENFELT, A., TAUCHID, M., XIE, X., GARRETT, R.G. & HALL, G.E.M., 1995. A Global Geochemical Database for Environmental and Resource Management. Recommendations for International Geochemical Mapping, Final Report of IGCP Project 259. UNESCO Publishing, 122 pp.

BØLVIKEN, B., BOGEN, J., JARTUN, M., LANGEDAL, M., OTTESEN, R.T. & VOLDEN, T., 2004. Overbank sediments: a natural bed blending sampling medium for large-scale geochemical mapping. Chemometrics and Intelligent Laboratory Systems, 74: 183-199. DOI: 10.1016/j.chemolab.2004.06.006.

de CARITAT, P. and COOPER, M.: National Geochemical Survey of Australia: Data Quality Assessment, Record, 2011/21, Geosci. Austral., Canberra, http://pid.geoscience.gov.au/dataset/ga/71971 (last access: 15 December 2022), 2011b.

It would be interesting to see how bulk soil 87/86Sr compares to 87/86Sr in other environmental materials within each catchment. I am interested in how it might compare to bioavailable 87/86Sr

leachates from soils for provenance studies. This is an idea for future work, rather than for inclusion in the current paper.

Thank you for the comment. We are working on acquiring complementary data for precisely such comparison.

Catchments with high 87/86Sr were compared to lithologies. It would be useful to see this comparison across all catchments.

We have added comparisons with carbonate and mafic lithologies, as also requested by Reviewer 2.

**Technical comments:**

Well structured, well written, easy to follow.

Thank you for the comment.

Figure 2a: Valuable diagram, but hard to make out bedrock units. Maybe adjust colour scheme and increase size?

The colours are standard geological colours so we prefer not to change these. We leave it to the Editor and the production team to ensure Figures are of an appropriate size.

Figure 4: May be worth including some of the major landmarks that you use in the discussion, for example, Cape York.

We feel that the Figures (maps) are already quite 'busy' so despite this being a great suggestion, fear that this could overload them. Also if you label Cape York, where do you stop?

Lines: 262 – 265: good points.

Thank you for the comment.

Table 1: Whilst digestion does hint at whether analyses were bulk soil versus bioavailable, I think it would be worthwhile making a clear distinction in the table which are which.

Whilst this is a great idea, we prefer not to enter the debate of what is 'bioavailable', as it is outside the scope of the paper. Some researchers will say it's only biological samples that represent bioavailable Sr, whilst others will accept weak/partial soil leaches. As the table is only for soils, the former definition is not applicable here.

Line 283: There is a good correlation between TOS and BOS 87/86Sr. Will you please explain why you would want to recalculate TOS for BOS?

We had TOS only available for some of the samples, but the majority were BOS. Therefore we needed a conversion of TOS to BOS as to not mix 'apples and oranges'. Some researchers interested in surface processes may prefer to use the TOS values. Therefore we provided the conversion between the two. We added the following sentence in Subsection 5.2:

*A TOS map and values would be useful when studying surface processes, such as plant and animal uptake or the effect of landuse on sediment composition and dynamics.*

Line 302-305: It would be great to compare this with all catchments and graph it similarly to Figure 6.

The relationship between Rb and 87Sr/86Sr is shown in the diagram below, which we don't think is clear enough to insert in the manuscript or supplement per se. It does however support the

statement made in the paper, namely that catchments with high 87Sr/86Sr values contain lithologies enriched in Rb relative to Sr

[Figure]

**Figure: 87Sr/86Sr isotope ratio (total digestion) vs Rb/Sr ratio (total digestion) for northern Australia NGSA BOS samples. Data subseted by geological age groups, as per Legend inset (note subsets with two or fewer samples not shown).**